# Critical Care Cycling Study (CYCLIST) trial protocol: a randomised controlled trial of usual care plus additional in-bed cycling sessions versus usual care in the critically ill

Marc R Nickels,[1] Leanne M Aitken,[2,3] James Walsham,[4,5] Adrian G Barnett,[6] Steven M McPhail[6,7]

For numbered affiliations see end of article.

**Correspondence to**
Marc R Nickels;
marc.nickels@health.qld.gov.au

## ABSTRACT

**Introduction** In-bed cycling with patients with critical illness has been shown to be safe and feasible, and improves physical function outcomes at hospital discharge. The effects of early in-bed cycling on reducing the rate of skeletal muscle atrophy, and associations with physical and cognitive function are unknown.

**Methods and analysis** A single-centre randomised controlled trial in a mixed medical-surgical intensive care unit (ICU) will be conducted. Adult patients (n=68) who are expected to be mechanically ventilated for more than 48 hours and remain in ICU for a further 48 hours from recruitment will be randomly allocated into either (1) a usual care group or (2) a group that receives usual care and additional in-bed cycling sessions. The primary outcome is change in rectus femoris cross-sectional area at day 10 in comparison to baseline measured by blinded assessors. Secondary outcome measures include muscle strength, incidence of ICU-acquired weakness, handgrip strength, time to achieve functional milestones (sitting out of bed, walking), Functional Status Score in ICU, ICU Mobility Scale, 6 min walk test 1 week post-ICU discharge, incidence of delirium and quality of life (EuroQol Five Dimensions questionnaire Five Levels scale). Quality of life assessments will be conducted post-ICU admission at day 10, 3 and 6 months after acute hospital discharge. Participants in the intervention group will complete an acceptability of intervention questionnaire.

**Ethics and dissemination** Appropriate ethical approval from Metro South Health Human Research Ethics Committee has been attained. Results will be published in peer-reviewed publications and presented at scientific conferences to assist planning of future multicentre randomised controlled trials (if indicated) that will test in-bed cycling as an intervention to improve the physical, cognitive and health-related quality of life outcomes of patients with critical illness.

**Trial registration number** This trial has been prospectively registered on the Australian and New Zealand Clinical Trial Registry (ACTRN12616000948493); Pre-results.

## Strengths and limitations of this study

► The randomised trial design with blinded assessments of skeletal muscle size, strength and function to provide objective measures of difference.
► The inclusion of an acceptability questionnaire will provide useful insights for subsequent implementation (if indicated).
► The study may not be powered for all secondary outcomes, and evidence of effect size from pilot data is not available for those measures.

## BACKGROUND AND RATIONALE

Patients with critical illness often require mechanical ventilation for periods greater than 48 hours. It has been identified that skeletal muscle wasting occurs early and rapidly during the first week of critical illness.[1] Despite international recommendations for patients with critical illness to commence activity as early as possible,[2] it has been identified that exercise interventions are rarely initiated when a patient is on mechanical ventilation.[3] This leads to prolonged immobility and may contribute to the development of intensive care unit-acquired weakness (ICUAW).[4] In-bed cycling using a cycle ergometer has been proposed as a safe and feasible method of introducing early exercise for patients with critical illness on mechanical ventilation who are sedated and immobile, this includes patients requiring inotropic support.[5–7] In-bed cycling may also assist in the preservation of muscle architecture. To date, only one randomised controlled trial (RCT) utilising in-bed cycling in the critically ill population has been published.[8] This single-centre RCT (n=90) conducted in Belgium found in-bed cycling to be safe for patients with critical illness and to improve

their 6 min walk distance, quadriceps force and Short Form-36 physical function scores on hospital discharge.[8] A limitation of this study was that the effects of the intervention at the muscular level were not assessed with muscle biopsy or ultrasound.[8] A pilot case-matched study of in-bed functional electrical stimulated cycling intervention in addition to usual care found positive physical outcomes observed among the cycling group including less time required to achieve functional milestones, time to stand and time to ambulate independently.[9] In addition, a shorter duration of delirium among those who participated in the functional electrical stimulation (FES) in-bed cycling intervention was observed.[9]

A recent clinical trial has demonstrated that patients with critical illness may experience persistent weakness despite participating in intensive exercise programmes while they are critically ill.[10] It has been suggested that intensive exercise programmes may not be effective if the commencement of these programmes is delayed.[11] Early exercise commencement is intended to assist in the maintenance of muscle mass. This may be achieved through a moderation of the inflammatory process.[11] Consequently, the effectiveness of exercise interventions that can commence early during critical illness are necessary to demonstrate if patient outcomes are improved by the early commencement of exercise during a period of critical illness.

Early clinical studies in the field have demonstrated potential for in-bed cycling interventions (with and without FES) to improve physical and cognitive function among patients with critical illness.[8 9] There are currently no published RCTs investigating the effectiveness of in-bed cycling in patients with critical illness requiring prolonged mechanical ventilation on quadriceps structure, ICUAW and cognitive outcomes. Consequently, further investigation of the effect of in-bed cycling on muscle structure, physical function and cognitive function is warranted.

## Objectives
The objectives of this study are to:
1. examine whether in-bed cycling in addition to usual care is effective in reducing the rate of rectus femoris cross-sectional area (CSA) atrophy and ICUAW in patients requiring more than 48 hours of invasive mechanical ventilation compared with usual care;
2. investigate if in-bed cycling in addition to usual care is associated with better functional and cognitive outcomes in patients predicted to require more than 48 hours of invasive mechanical ventilation compared with usual care.

## METHODS
### Study design and setting
This trial will be a two-arm, parallel RCT with individual participant allocation and blinding of the primary outcome assessor. It will be conducted in a 25-bed tertiary mixed medical and surgical adult intensive care unit (ICU) in Brisbane, Australia. Participants will be allocated 1:1 to receive either usual care or in-bed cycling in addition to usual care (figure 1). In designing this study, the[1] SPIRIT 2013 Checklist was used to ensure that all recommended items in a clinical trial were addressed.[12]

### Consent
It is anticipated that most patients who are eligible to participate in this study in the ICU setting will not be able to provide informed consent at the time of study enrolment. For those patients who may have the capacity to provide informed consent, the Richmond Agitation and Sedation Scale (RASS) will be used to determine if a patient is rated 'Alert and Calm'. If a patient is rated as 'Alert and Calm' on the RASS, the Confusion Assessment Method for the Intensive Care Unit (CAM-ICU) will be used to determine if a patient has had delirium within the preceding 24 hours. Provided a patient passes the RASS and CAM-ICU assessments, the treating clinical team will be approached to determine if the patient has the capacity to provide informed consent. Patients without delirium for the preceding 24 hours and deemed to have capacity will be approached to provide their own written informed consent for study participation. For eligible patients considered unable to provide informed consent at the time of study enrolment, substitute decision-makers (family members or next of kin) will be approached for written informed consent. The Queensland Civil and Administrative Tribunal (QCAT) has approved an application to provide consent for individuals who are unable to give consent and do not have a next of kin who is accessible to request consent on the patients' behalf. Delayed consent from the patient will be sought once they can provide consent for themselves if they are enrolled using QCAT approval. Participation in the study is voluntary. Patients or their substitute decision-makers are able to withdraw at any time without any negative consequences on the care they would receive, their ongoing relationship with the hospital, or staff involved in their care.

The substitute decision-maker consent form (PICF_SDM_CYCLIST_V3.0_20160701) is attached as an online supplementary file 1.

### Randomisation
Patients will be individually randomised in a 1:1 ratio to either intervention or usual care group. Blocking (random block sizes) will be used to help balance the groups. An investigator not involved in the screening, consenting, allocation or assessment processes will use computerised random number generation to create the randomisation sequence. A randomisation sequence will be uploaded onto the Research Electronic Data Capture (REDCap) secure web-based computer application.[13] The REDCap randomisation module will reveal the group allocation of each patient to the intervention coordinator after a patient's baseline data have been collected.

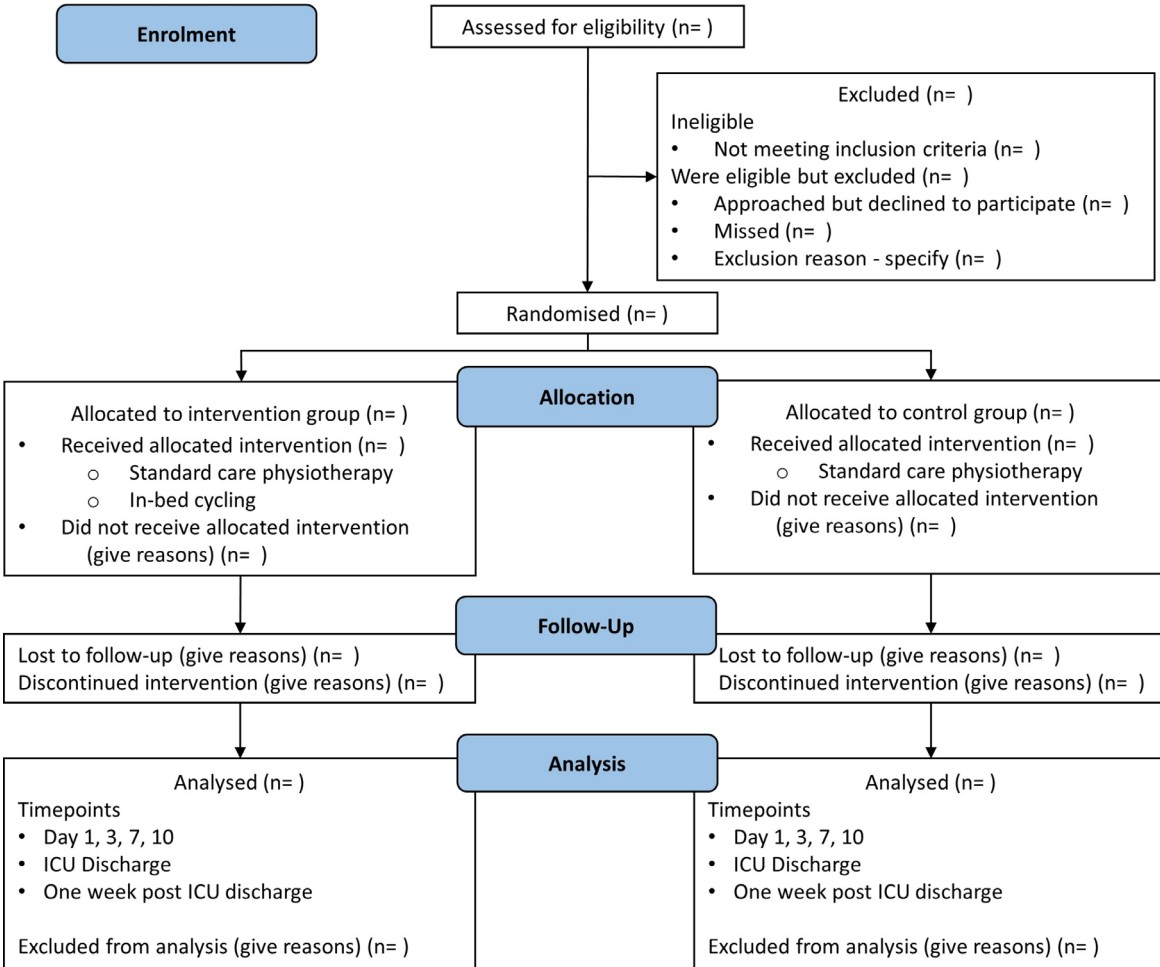

**Figure 1** Consort diagram giving flow of participants throughout the study. ICU, intensive care unit; n, number.

## Participants

The study aims to recruit 68 participants. Adult patients expected to require at least 48 hours of mechanical ventilation will be recruited, with the inclusion and exclusion criteria listed in table 1. Criteria to guide when to discontinue or not deliver an intervention are listed in box 1. Participants can be recruited into the study and baseline sonography measures performed if a patient does not

| Table 1 | Inclusion and exclusion criteria |
|---|---|
| **Inclusion** | **Exclusion** |
| Expected to require more than 48 hours of invasive mechanical ventilation | Pre-existing condition that is likely to impair mobility/mobility assessment (ie, significant neurological, musculoskeletal, cognitive or mental health disorder) |
| Able to provide consent or have a family member consent on their behalf | Neuromuscular disorder or acute primary brain lesion (ie, traumatic brain injury, intracranial haemorrhage, stroke or hypoxic brain injury) |
| Enrolled into the study within 96 hours of ICU admission | Injuries precluding cycle ergometry (ie, spinal/pelvic/lower limb orthopaedic injuries/open abdominal wound) |
| Expected to remain in ICU for more than 48 hours following study enrolment | Obesity >135 kg (MOTOmed Letto 2 maximum rated weight capacity) |
| | Uncontrolled seizures or status epilepticus |
| | Dire prognosis (ie, unlikely to survive the current admission) |
| | Pregnancy |
| | Children and/or young people (ie, <18 years) |

ICU, intensive care unit.

## Box 1    Safety guidelines

*Active or passive exercise should not be delivered if:*
► Clinician's opinion that patient's condition is unstable
► Resting HR<40 or >120 bpm or new arrhythmia
► Evidence of coronary ischaemia, for example, chest pain or ECG changes
► MAP<60 or SBP>200 mm Hg
► $SpO_2$<90%
► RASS≥2
► Wounds of leg, pelvis or lumbar spine precluding cycle ergometry
► Evidence of active bleeding or coagulation disorder: INR>1.8, PLT<50 000/µL*
► Femoral vascular access†, for example, dialysis catheter, IABP, ECMO or lower limb arterial line‡
► Acute DVT or PE

*Active exercise should not be delivered if:*
► >20 µg/min of norepinephrine or comparable inotropic or vasopressor support
► $FiO_2$>0.55 or PEEP>10 cmH2O
► RR>30 with adequate ventilatory support
► Temperature >39°C

*Stopping criteria: active or passive exercise should cease if:*
► HR<50 or >140 bpm, or new arrhythmia develops (including ventricular ectopic or new-onset AF)
► Evidence of coronary ischaemia, for example, chest pain or ECG changes
► MAP<60 mm Hg
► SBP>200 mm Hg
► Clinical signs of cardiorespiratory distress
► $SpO_2$<90% for more than 1 min
► Patient's request to stop therapy

*Values outside this range would be tolerated if patient is therapeutically anticoagulated.
†Other than femoral central line.
‡If a femoral vascular access is inserted unilaterally the contralateral leg may be cycled unilaterally.
AF, atrial fibrillation; bpm, beats per minute; DVT, deep vein thrombosis; ECMO, extracorporeal membrane oxygenation; $FiO_2$, fraction of inspired oxygen; HR, heart rate; IABP, intra-aortic balloon pump; INR, international normalised ratio; MAP, mean arterial pressure; PE, pulmonary embolism; PEEP, positive end expiratory pressure; PLT, platelets; RASS, Richmond Agitation Sedation Score; RR, respiratory rate; SBP, systolic blood pressure; $SpO_2$, saturation of peripheral oxygen.

currently meet the in-bed cycling safety criteria in box 1 (but are considered by their treating clinical team to be likely to meet the criteria within their stay in ICU).

### Study intervention

All study patients will receive usual physiotherapy interventions while in intensive care. Physiotherapy interventions will include (but are not limited to): respiratory physiotherapy, physical rehabilitation exercise interventions including sitting on the edge of the bed, sit to stand transfers, sitting out of bed and walking. Usual care physiotherapy interventions will be prioritised over the in-bed cycling intervention. Safety guidelines (see box 1) will be used to determine if the intervention group patients are able to complete an additional daily 30 min progressive lower limb in-bed cycling using a bedside cycle ergometer

(MOTOmed Letto 2). The lead physiotherapist (MRN) who has over 10 years of experience in rehabilitative exercise with patients with critical illness will primarily conduct the majority of in-bed cycling sessions. He has been trained by industry cycle ergometry representatives and has over 5 years of experience conducting in-bed cycling sessions with patients with critical illness. Experienced ICU physiotherapists trained in conducting in-bed cycling sessions may conduct the in-bed cycling sessions if the lead physiotherapist is unavailable. During the in-bed cycling exercise interventions, the patients' vital signs will be monitored. If the intervention group patients are in a state of low arousal or sedated, they will cycle continuously and passively at the default passive speed of the cycle ergometer (20 revolutions per minute) for 30 min. When the participant is following commands, the clinician will verbally encourage the patient to complete in-bed cycling sessions actively for a duration of 30 min. Once the patient can cycle actively, the resistance applied by the cycle ergometer will be adjusted to facilitate patient intensity between 3 and 5 using the visual Borg scale rate of perceived exertion (category ratio 10),[14] within the specified safety guidelines. An exercise intensity of 3–5 on the Borg rate of perceived exertion scale has been shown to be safe and feasible with patients with critical illness.[15] This will enable the patient to cycle in-bed either passively or actively with assistance from the cycle ergometer. If a patient unexpectedly commences active cycling, the additional active in-bed cycling safety criteria will become relevant. If the patients are deemed unsuitable to continue active in-bed cycling, they will be asked to resume passive cycling. If the patient continues to actively cycle, the session will be ceased. The in-bed cycling sessions will continue until the patient completes a minimum of five in-bed cycling sessions, unless the patient is discharged from hospital prior to completing five sessions. The intervention will continue in the acute hospital ward if the patient is discharged from ICU prior to completing five in-bed cycling sessions. While the patient remains in ICU, in-bed cycling sessions will continue (up to 7 days/week), up to 28 days post-ICU admission. This frequency is congruent with usual physiotherapy services that can provide rehabilitative exercise interventions to ICU patients up to 7 days/week. Patients randomised to the usual care arm do not routinely complete in-bed cycling sessions during their hospitalisation. Any deviations from the planned protocol will be recorded to enable appropriate intervention description and, if indicated, a per-protocol analysis (in addition to the primary intention-to-treat analysis).

Participants will not be coerced to complete any intervention or outcome measure. Participants may be discharged home from the participating acute hospital before they have completed outcome measures at each assessment time point. If this occurs, participants will be asked to return to the hospital to enable the remaining outcome measures to be completed and expenses related to taxi or parking costs will be reimbursed.

**Table 2** Descriptions of outcome measures for CYCLIST RCT

| Assessment component | Outcome measure | Description |
|---|---|---|
| Muscle morphology | Ultrasound | RF CSA, AP thickness of RF and VI. Measured in triplicate on right anterior thigh one-third distance from superior patella to ASIS. Patient positioned in supine, 30° head elevation[27] |
| Muscle strength | MRC sum score | Standardised sum of 12 MMTs, 3 MMTs per limb Score ≤48 indicative of ICU-acquired weakness[28] |
| | Handgrip strength dynamometry | Triplicate bilateral measurement using a Jamar Digital Dynamometer (Lafayette) with seated patient[29] |
| Physical function | ICU Mobility Scale | Best level of function achieved in ICU using an 11-point ordinal scale[30] |
| | FSS-ICU | Patients' function measured an 8-point ordinal scale[31 32] |
| | Functional milestones | Time to achieve functional milestones: sit out of bed, time to stand, mobilise with assistance and mobilise independently |
| | 6 min walk test | Submaximal endurance test of distance walked by a patient in 6 min[33] |
| Cognition | CAM-ICU | Incidence and recorded episodes of acute delirium[34] |
| Quality of life | EQ-5D-5L | EQ-5D-5L[35] |
| Intervention acceptability | Customised questionnaires | Questionnaire about the acceptability of the in-bed cycling intervention |

AP, anterior posterior; ASIS, anterior superior iliac crest; CAM-ICU, Confusion Assessment Method for the Intensive Care Unit; CSA, cross-sectional area; CYCLIST, Critical Care Cycling Study; EQ-5D-5L, EuroQol Five Dimensions questionnaire Five Levels scale; FSS-ICU, Functional Status Score for the Intensive Care Unit; ICU, intensive care unit; MMT, manual muscle test; MRC, Medical Research Council; RCT, randomised controlled trial; RF, rectus femoris; VI, vastus intermedius.

## Outcomes

Table 2 provides a summary of the outcome measures. The primary outcome is the percentage of change in rectus femoris CSA measured at baseline (within 24 hours of study enrolment) and day 10 (post study enrolment), measured by a blinded assessor. The participants will be assessed to examine whether there is a between-group difference in the amount of rectus femoris CSA atrophy. The authors anticipate that less rectus femoris CSA atrophy will be observed among the in-bed cycling group. Day 10 post study enrolment has been chosen as the primary endpoint to enable comparison with previously published data on acute skeletal muscle wasting in patients with critical illness and consistent with this previously reported time frame of observed muscle wasting.[1] Secondary outcome measures in the CYCLIST (Critical Care Cycling Study) are muscle strength, physical function, cognition, quality of life and acceptability of intervention.

Demographic information such as age, gender and diagnostic code will be collected. Illness-related information including length of mechanical ventilation, ICU length of stay (LOS)/hospital LOS and discharge destination, illness severity (Acute Physiology and Chronic Health Evaluation III,[16] Sequential Organ Failure Assessment[17] score), premorbid comorbidities, neuromuscular blockade and sedation medications administered, nutrition received, cumulative fluid balance, patient's height and weight, and body mass index will also be collected.

## Sample size

Based on the repeated measures design, this study will have 80% power to detect a between-group difference of 2.9% on our primary outcome (change in rectus femoris CSA) between the intervention and usual care participants (assuming type I error 0.05, SD of 6%, and within-patient correlation of 0.5 between assessments, and after accounting for an up to 20% dropout rate including in-hospital mortality). A total of 68 patients (34 in each group) will be recruited into the study. With a dropout rate of up to 20%, this would represent approximately 27 participants completing the trial in each group. It is noteworthy that some of the assumptions for this calculation have been made in the absence of prior RCT data. There is some risk that the observed trial data may not fit these assumptions (eg, discrepancies in SD or within-patient correlation between the repeated measures), and the power of the study may differ from this a priori sample size estimate.

## Data collection

The study data will be collected at the time points summarised in table 3.

The management of study data will be dependent on the type of data collected. Baseline data will be entered directly into REDCap-designed digital clinical trial workflow management software.[13] Ultrasound results will be uploaded onto the secure hospital-based AGFA IMPAX 6.5.3.1005 Medical Image Viewer application. In-bed cycling session data, physical assessment measures,

**Table 3** CYCLIST summary of time points of assessments

| | Baseline | Day 3 | Day 7 | Day 10 | ICU discharge | 1 week post ICU discharge* | 3, 6 months' post ICU-admission |
|---|---|---|---|---|---|---|---|
| **Severity of illness** | | | | | | | |
| APACHE III | ✓ | | | | | | |
| SOFA | ✓ | ✓ | ✓ | ✓ | ✓ | | |
| **Muscle morphology** | | | | | | | |
| Quadriceps ultrasonography | ✓ | ✓ | ✓ | ✓ | | ✓ | |
| **Strength measures** | | | | | | | |
| MRC sum score | | | | | ✓ | ✓ | |
| Handgrip dynamometry | | | | | ✓ | ✓ | |
| **Physical function measures** | | | | | | | |
| ICU mobility scale | | | | | ✓ | | |
| FSS-ICU | | | | | ✓ | | |
| Functional milestones | ✓ | ✓ | ✓ | ✓ | ✓ | ✓ | |
| 6-MWT | | | | | | ✓ | |
| **Cognition** | | | | | | | |
| CAM-ICU | ✓ | ✓ | ✓ | ✓ | ✓ | | |
| **Quality of life** | | | | | | | |
| EQ-5D-5L | | | | | ✓ | | ✓ |
| **Acceptability of intervention** | | | | | | | |
| Participant acceptability | | | | ✓† | | | |
| **Patient outcomes** | | | | | | | |
| ICU length of stay | | | | | ✓ | | |
| Hospital length of stay | | | | | | ✓‡ | |
| Acute discharge destination | | | | | | ✓‡ | |
| Mortality | | | | | ✓ | ✓‡ | ✓ |

*1-week post-ICU discharge or at acute hospital discharge if sooner.
†At completion of in-bed cycle ergometry sessions.
‡Measured at acute hospital discharge.
6-MWT, 6 min walk test; APACHE, Acute Physiology and Chronic Health Evaluation; CAM-ICU, Confusion Assessment Method for the Intensive Care Unit; CYCLIST, Critical Care Cycling Study; EQ-5D-5L, EuroQol Five Dimensions questionnaire Five Levels scale.; FSS-ICU, Functional Status Score for the Intensive Care Unit; ICU, intensive care unit; MRC, Medical Research Council; SOFA, Sequential Organ Failure Assessment score.

acceptability of in-bed cycling intervention and quality of life questionnaires will be recorded initially onto research data sheets. All information recorded on data sheets will be subsequently entered into the REDCap application.

Assessments of muscle size will be assessed by ultrasound and performed by registered postgraduate trained sonographers with expertise in musculoskeletal sonography. Sonographers will be blinded to patient treatment group allocation and will be responsible for analysing and scoring all ultrasound images. The study sonographers were involved in the development and standardisation of the ultrasound procedure prior to the commencement of the study.

Assessment of muscle strength and function will be completed by cardiorespiratory physiotherapy assessors with a minimum of 3 years' experience at participants' ICU discharge and 7 days post-ICU discharge. These physiotherapists will be blinded to group allocation. The same physiotherapists will also assess 6 min walk test distance 7 days following ICU discharge. The study physiotherapists were trained in the standardised assessment of the study outcome measures prior to the commencement of the study.

A physiotherapist not involved in blinded outcome assessment will be responsible for conducting the in-bed cycling sessions, and the remaining 'usual care' physiotherapy will be completed by hospital department physiotherapists not involved in the study. Additionally, the primary statistician will be blinded to treatment group allocation.

To minimise the chances of unintentional unblinding, at the beginning of each assessment the blinded assessors will clearly state that they wish to complete the assessment process without knowing which group the patient was allocated to. If any blinded assessors were to become unblinded to group allocation for any patient (eg, through inadvertent revelation by a patient being assessed or unanticipated exposure to a patient taking part in an in-bed cycling session), they have been instructed to report this to the intervention coordinator (MRN) who will also record when this occurred.

## Analysis

Data will be analysed and reported using intention-to-treat principles (primary analysis). A per-protocol analysis will be conducted to assist in determining the efficacy of the in-bed cycling protocol if variation from the planned protocol occurs for a substantial proportion of patients. The per-protocol analysis will include only patients who adhered to the protocol and received at least 80% of training sessions (minimum of four sessions).

Descriptive statistics and generalised linear mixed models (that can adjust for baseline status) will be used to examine the effect of group allocation (intervention vs usual care) on the primary and secondary outcomes.[18] As this is a randomised trial, we do not plan to adjust for potential confounders (eg, age, gender, comorbidities), but will compare the characteristics of the sample by treatment group and may adjust if a potential confounder differs greatly between groups.

If blinded sonographers are unable to complete an ultrasound measurement at designated time points, then the scan will be performed within a day (±1 day) of the designated time point. A subsequent sensitivity analysis will be conducted to determine if there is a difference by measurement timing. If the primary outcome cannot be collected by a sonographer within the specified time frame, a physiotherapist experienced in musculoskeletal ultrasound will measure thigh muscle size. The measurement will be from the most anterior aspect of rectus femoris to the anterior surface of the femur. This will assist in the imputation of the missing primary outcome data. Statistical analysis of available data will be used for the primary analysis when data are completely missing. Multiple imputation will be conducted if more than 20% of outcome data are missing. Additional planned secondary analyses are listed in box 2. The principal investigators will determine if study protocol modifications are required and any modifications to the existing protocol will be declared.

## Trial management

The principal investigator (MRN) will oversee the conduct and progress of the trial. The principal investigator will screen the daily admission to ICU lists and liaise with the treating medical teams to optimise participant enrolment. No interim analyses are planned. The principal investigator will ensure all research personnel are

---

### Box 2   Planned secondary analyses

- ► Muscle wasting at baseline, day 3, day 7 and 1 week post-ICU discharge
- ► Muscle wasting adjusted for number of failing vital organs*
- ► Muscle wasting adjusted for severity of illness on admission to ICU†
- ► Muscle wasting adjusted for the number of days prior to a patient commencing active activity
- ► Muscle wasting adjusted for the patients' cumulative fluid balance on the day of the ultrasound scan
- ► Relationship between muscle wasting and participants' nutritional intake while in ICU
- ► Relationship between sedative and paralytic medications and muscle wasting
- ► Cost comparison of hospitalisation for both the intervention and usual care groups‡
- ► Hospital readmissions after acute hospital discharge over a 2-year time period§
- ► Mortality¶

Utilising:
*Sequential Organ Failure Assessment (SOFA).
†APACHE III (Acute Physiology and Chronic Health Evaluation) data.
‡Health service utilisation data.
§Queensland Hospital Admitted Patient Data Collection (QHAPDC).
¶Queensland Health Statistical Services Branch.
ICU, intensive care unit.

---

appropriately orientated and trained, oversee recruitment and report to a trial safety monitoring committee who will monitor the progress and conduct of the trial. The trial safety monitoring committee will include: a physiotherapist and researcher experienced in the safe conduct of clinical trials with physiotherapy interventions; the trial coordinator and principal investigator; a critical care nurse who is also experienced in the safe conduct of clinical trials in critical care settings; and two ICU medical consultants experienced with the safe conduct of clinical trials in ICUs. One of the ICU medical consultants is employed externally to the study site. The principal investigator will provide an update report to the safety monitoring committee on a monthly basis (and additional ad hoc reports if an adverse event occurs). Additionally, any serious adverse events will be reported to the approving Metro South Human Research Ethics Committee that is overseeing the study. During in-bed cycling sessions, participants will be monitored for adverse events that will be recorded on the session data collection form. Adverse events that are being monitored for are: line or airway dislodgement, increase in ventilatory support that persists longer than 5 min post exercise (eg, increase in positive end expiratory pressure or fraction of inspired oxygen), blood oxygen desaturation less than 88% for more than 1 min, increase in vasoactive or pain relief medication greater than 5 mcg/min, increase in systolic blood pressure greater than 180 mm Hg for more than 2 min, increase in heart rate greater than 140 bpm for more than 2 min, decrease in mean arterial blood pressure less than 60 mm Hg for greater than 2 min

and decrease in heart rate less than 50 bpm for more than 2 min.

## Dissemination

Study results will be disseminated via publication in peer-reviewed literature and scientific conference presentations. It is anticipated that media releases in lay form will be completed to target the general community. Study results will also be placed on a university website for viewing by participants and other interested parties. There are no publication restrictions. Authorship eligibility guidelines as outlined in the Australian Code for the Responsible Conduct of Research[19] and consistent with those proposed by the International Committee of Medical Journal Editors will be followed to determine authorship.[20]

## DISCUSSION

Survival rates following critical illness are improving[21]; however, patients are experiencing deficits in physical and cognitive function that do not equal age-matched peers 5 years after an episode of critical illness.[22] The delayed initiation of rehabilitative exercise interventions with patients with critical illness may explain the limited effectiveness of clinical trials that have studied the effect of exercise interventions on patients' functional outcomes.[11] A binational clinical trial that aimed to commence exercise interventions as early as possible with mechanically ventilated patients reported that despite the presence of a dedicated early mobility team, patient mobilisation out of bed while mechanical ventilation was in situ was rare.[23] These results are substantiated by point prevalence studies from Australia, New Zealand and Germany that report that in 1281 patient days only one patient with an endotracheal tube was mobilised out of bed.[3 24] The presence of an endotracheal tube is negatively correlated with out-of-bed mobilisation in the USA with a reported OR of 0.1 (95% CI 0.05 to 0.2).[25] Studies have reported that following a period of critical illness, rehabilitative interventions do not hasten recovery when they are provided after acute hospital discharge.[15 26] Consequently, this trial will provide valuable clinical trial evidence regarding the effect of an exercise intervention initiated early in the illness of patients with critical illness. Specifically, it will report empirical data about the effect of the intervention on the rate of skeletal muscle wasting, and whether early exercise interventions that can be feasibly implemented among people who are mechanically ventilated are associated with improved physical, cognitive and health-related quality of life outcomes.

CYCLIST is a phase IIb RCT that is powered to investigate if the early application of an additional in-bed cycling intervention is able to reduce the rate of skeletal muscle atrophy of patients' quadriceps muscle during and immediately following a period of critical illness, in comparison to usual care. Secondary outcomes evaluated in this study will assist sample size calculations for future studies to assess for efficacy of functional outcomes. The study will also aid planning of future rehabilitation-based clinical trials with regard to rate of participant recruitment. The investigators are planning to meet at the completion of the study to facilitate reflection on aspects of the study that could be improved and to consider whether a definitive phase III RCT is warranted. This will include consideration of the evidence of effect on primary and secondary outcomes, participant recruitment and adherence to the in-bed cycling protocol and rate, as well as acceptability of the intervention.

The strengths of this study are the implementation of the in-bed cycling intervention as soon as feasible (including while patients are still mechanically ventilated), with clear commencement and stopping rules. Another strength of the study is the blinded assessment of muscle structure, strength and function assessments. Measurement of the effect of early in-bed cycling on quality of life up to six months post ICU-admission is another strength that will assist to demonstrate if potential gains made early during a period of critical illness correspond to lasting functional improvements. It is expected that this study will also provide insights regarding the feasibility of the in-bed cycling intervention with patients with critical illness from the rates of compliance and completion of the in-bed cycling exercise intervention. The acceptability of in-bed cycling intervention from the perspective of patients with critical illness will be sought through a questionnaire and provide new information to inform potential implementation strategies for this intervention. A potential limitation of the study may be difficulty completing functional assessment measures at ICU discharge and at 1 week post-ICU discharge with patients who are either profoundly weak or present with an acute cognitive dysfunction. Further, usual care physiotherapy interventions will be delivered by treating teams independent of the study and prioritised over in-bed cycling. The frequency, intensity, type and duration of all usual care physiotherapy interventions are not being prospectively recorded as part of the trial outcomes. As the study design is an RCT, it is anticipated that the usual care physiotherapy will be similar across groups. Also this study is being conducted at a single centre and therefore results may need to be interpreted with caution.

**Author affiliations**
[1]Department of Physiotherapy, Princess Alexandra Hospital, Woolloongabba, Queensland, Australia
[2]School of Health Sciences, City, University of London, London, UK
[3]National Centre of Research Excellence in Nursing Interventions for Hospitalised Patients, Menzies Health Institute Queensland, Griffith University, Brisbane, Queensland, Australia
[4]Intensive Care Unit, Princess Alexandra Hospital, Woolloongabba, Queensland, Australia
[5]University of Queensland, Brisbane, Queensland, Australia
[6]Institute of Health and Biomedical Innovation, Queensland University of Technology, Brisbane, Queensland, Australia
[7]Centre for Functioning and Health Research, Metro South Hospital and Health Service, Brisbane, Queensland, Australia

**Collaborators** The principal investigator MRN will have full access to the final trial dataset. Associate investigators SMM, LMA, JW and AGB will have access to the final de-identified trial dataset. Other investigators will be granted access to sections of the final trial dataset. Dataset sections may include de-identified demographic data and information that pertains to data collected by the professional discipline of the associate investigator.

**Contributors** MRN contributed to study conception, design, grant acquisition, trial management (including intervention coordination, data collection), data management, protocol drafting, appraisal and editing. SMM, LMA and JW contributed to study conception, design, grant acquisition, analysis plan, data management, protocol drafting, appraisal and editing. AGB contributed to study analysis plan, grant acquisition, data management, protocol drafting, appraisal and editing. All authors read and approved the final manuscript.

**Funding** This is an investigator-initiated trial without external sponsors. Following a competitive peer review process Metro South Health Study, Education and Research Trust Account (SERTA) awarded this study a grant in 2015. MRN has also been awarded a competitive Princess Alexandra Research Support Scheme Postgraduate Scholarship to conduct this study. SMM (#1090440) and AGB (#1117784) are supported by National Health and Medical Research Council (NHMRC) fellowships. In addition, this study is receiving in-kind support in the form of personnel and administrative support from Metro South Health (Queensland) to enable the study to be conducted. No funding body had a role in study design, collection, management, analysis and interpretation of data; writing of the report; and the decision to submit the report for publication.

**Competing interests** None declared.

**Ethics approval** Human research ethics approvals for this study have been gained from Metro South Human Research Ethics Committee (EC00167) on 28 April 2016 (HREC/16/QPAC/193), and subsequent approval following an administrative review from Queensland University of Technology Human Research Ethics Committee (QUT reference number: 1600000441). Ethics approval has been granted until 28 April 2019. Site-specific approval (SSA) has been granted by Metro South Centres for Health Research, Research Governance (SSA/16/QPAH/195) on 1 June 2016. CYCLIST Study Protocol Version 2.1 dated 30 March 2017 was approved on 13 April 2017. Participant recruitment commenced on 26 July 2016.

**Provenance and peer review** Not commissioned; externally peer reviewed.

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
