## [Reviewer comments · BMJ Open]

ARTICLE DETAILS

TITLE (PROVISIONAL)	Critical Care Cycling Study (CYCLIST) Trial Protocol: a randomised controlled trial of usual care versus usual care plus additional in-bed cycling sessions in the critically ill
AUTHORS	Nickels, Marc; Aitken, Leanne M.; Walsham, James; Barnett, Adrian; McPhail, Steven

VERSION 1 – REVIEW

REVIEWER	James M. Smith, DPT, MA Professor of Physical Therapy Utica College Utica, NY, USA
REVIEW RETURNED	08-May-2017

GENERAL COMMENTS	While much evidence has led to recommendations for early mobilization / early rehabilitation interventions for patients in the intensive care unit (ICU) there continues to be a need for evidence on interventions, and intervention dosing, for those patients. This study presents an effective proposal for evaluating one strategy for the management of exercise for these patients. The design of this study should be appropriate for meeting the objectives described on p. 5, lines 4 – 14, and that will expand the knowledge about managing this fragile patient population. I found that this protocol article read well and made sense. This study should extend on what is already in the published literature, as was summarized on p. 4, lines 45 – 53. That information should prove important for critical care practitioners, especially those who provide rehabilitation services. Note that the claim on p. 4, lines 45 – 48 will benefit from a citation. My concern, as identified in the review checklist, is “4. Are the methods described sufficiently to allow the study to be repeated?” Rehabilitation interventions are notorious for being insufficiently described in the literature. The interventions in this protocol are described sufficiently for the study to be repeated except for the intensity of the exercise intervention. That is, my primary concern here is the planned “adjustment” of the ergometer “to facilitate optimal patient intensity.” As described in the TIDieR checklist- “Tailoring: If the intervention was planned to be personalised, titrated or adapted, then describe what, why, when, and how” (from Hoffmann TC, et al. Better reporting of interventions: template for intervention description and replication [TIDieR] checklist and guide.
--

	BMJ 2014;348:g1687 doi: 10.1136/bmj.g1687, p. 6). It appears that this intervention will be titrated to the subjects' response. I recommend that elaboration on how intensity will be determined, and adjusted, will be informative for the provision of the intervention and for the replication of this study. I also have concern about the exclusion criteria, as described in Table 1 (p. 6). I agree with the need for exclusion of subjects with a pre-existing condition that is likely to impair mobility / mobility assessment. It may improve clarity to change from examples for which subjects will be excluded (e.g., line 33) to specifics for which subjects will be excluded (that is, consider changing "e.g." to "i.e."). A similar concern applies to the descriptor for dire prognosis that is identified on line 43. Beyond that my observations are rather petty, such as spelling errors (e.g., morophology, p. 8, line 39; and, p. 14, line 28) that will be resolved in the editing process. I found the overall design to be appropriate for a complex environment with at-risk and fragile subjects. The authors have described a reasonable strategy to enroll and to protect the subjects while engaging them in this research. The safety guidelines (table 2, p. 7 - 8) are consistent with standards for protecting patients participating in early rehabilitation in the ICU. Thank you for this opportunity to review this proposal and I hope my review is informative.
--	--

REVIEWER	Zudin Puthuchery Royal Brompton Hospital United Kingdom
REVIEW RETURNED	11-May-2017

GENERAL COMMENTS	Nickels et al propose a phase IIb randomised controlled trial of in-bed cycling for the primary prevention of muscle wasting in critically ill patients. This addresses a highly relevant gap in the literature. Whilst the study methodology is sound, i have a few comments that i hope will be taken in the spirit they are offered- to improve study design. 1) The time has passed for unselected patients entering rehabilitation or prevention trials in critical care. Instead we are searching for more nuanced work. Publications from Herridge et al, Puthuchery et al and upcoming publications from the RECOVER trial demonstrate the variable responses of our patients to interventions, and i would suggest you either stratify patients for potential response, or adjust for baseline. This may involve retrospective HRQOL via Families, CPAT, functional histories (e.g. Ferrante JAMA int med 2015) or clinical frailty scoring. At least 2-3 trials have been skewed as result of not adjusting for baseline. 2) You have not stated that you will collect nutritional data- exercise and nutrition being intertwined in muscle physiology (see Bear Lancet Resp 2014). Given point 1 above i would collect data for NUTRIC scoring as this is likely to predict response to some extent 3) Are femoral lines contraindicated? Hodgkins has published consensus on mobilisation, and Parry's FES pilot study did not use this as an exclusion criteria.
---

	4) How will the change in muscle mass be assessed statistically ? This may work better with a RM-ANOVA as opposed to t-test. 5) 20% missing data would be a worrying level- Imputation would not be appropriate at >10%
--	--

REVIEWER	Michelle Kho McMaster University, Canada
REVIEW RETURNED	25-May-2017

GENERAL COMMENTS	1 Critical Care Cycling Study (CYCLIST) Trial Protocol: a randomised controlled trial of usual care versus usual care plus additional in-bed cycling sessions in the critically ill Nickels et al. describe a research protocol for in-bed cycling with critically ill patients. It is a single-center, parallel group randomized trial with blinded outcome measures. Patients will be randomized to 30 minutes of cycling and routine physiotherapy interventions or routine physiotherapy interventions alone for up to 28 days, with a minimum of 5 cycling sessions whilst in hospital (ICU and acute ward combined). The primary outcome is change in rectus femoris cross-sectional area compared to baseline at 10 days post-study enrolment. Secondary outcomes include strength (MRC sum score, hand grip), function (ICU mobility scale, functional status score for ICU, 6-minute walk test), cognition (CAM-ICU up to 10 days post-enrollment), quality of life (EQ5DL 10 days post-enrolment, 3 and 6 months), ICU and hospital LOS, acute discharge destination, and mortality. Congratulations to the investigators on initiating an early randomized study of in-bed cycling with critically ill patients. We need more rigorous research in this field, and it is great to have others studying interventions to help our ICU survivors. The authors include some strong design features to ensure intervention exposure (e.g., minimum 5 cycling sessions), patient acceptability, and use common ICU outcomes. Given this study is already enrolling, I will focus on areas for clarification and better research reporting. Thanks for including the SPIRIT checklist. Main issues: 1. Primary outcome – The primary outcome is change in rectus femoris quadriceps cross sectional area measured at 10 days post-enrollment compared to baseline. The authors will enroll 68 patients to demonstrate a 2.9% difference in change between groups. a. Rationale for primary outcome: i. More information to justify the choice of this outcome, rather than a functional or patient-centered outcome is needed, since completed or ongoing trials
--

	reported outcomes of the 6MWT or feasibility. What is the clinical relevance of muscle cross sectional area, rather than function? ii. Information about the rationale for the timing of this outcome (i.e., 10 days) is needed. iii. Please clarify the direction of change (e.g., the cycling group will have less change in CSA than usual care PT). iv. A 2.9% difference in change score between groups is small. What is the expected precision of the measure between groups (i.e., 95% confidence interval)? The study cited to support the sample size calculation reported a mean change of -17.7% (-20.9% to -4.8%), representing a wide confidence interval for the control group. v. Please clarify how many assessments are required per group (you will recruit 68 to account for 20% drop out rate). How does this sample size calculation account for ICU mortality? b. Assessment of primary outcome: Please provide more information about the expertise and training provided to the blinded outcome assessor. c. Missed outcomes: On page 11, line 36, the authors report that if sonographers do not record the primary outcome, then a physiotherapist will use thigh circumference measures to impute missing outcome data. Please provide rationale for this statement, given poor psychometrics of thigh circumference measures (Parry et al., Intensive Care Med (2015) 41:744–762). Alternatively, would the authors consider adding a time window to conduct the 10-day ultrasound measure (e.g., +/- 1 day) and conduct a sensitivity analysis to see if there is a difference by measurement timing? 2. Study intervention: Some elements of the cycling intervention are not clear. Can the authors please clarify the following: a. At the time of enrolment, do patients need to meet the safety criteria on page 7, table 2? How will you manage patients enrolled in the study, randomized to the cycling arm, and don't receive the treatment? b. Will this study prioritize routine physiotherapy interventions over in-bed cycling (see page 6, line 56)? The authors report "safety guidelines will be used to determine if the intervention group patients are able to complete an additional daily 30 minute...." c. Please justify why the initial cycling rate is 20 RPM, and the rationale for adding resistance. d. Please clarify the number of days per week cycling is offered as
--	--

	part of the study (e.g., 5 days per week, 7 days per week, other). Is this frequency of intervention similar to current ICU PT scheduling? e. Given additional safety criteria for active cycling, how will the authors manage patients who transition from passive to active cycling in the same session and have an active exercise exclusion? f. Please provide more information about the expertise and training provided to the interventional physiotherapist for cycling. 3. Control group: How will authors document the frequency, intensity, type, and duration of physiotherapy interventions in the control group? Other issues: 1. Secondary outcomes: Please provide more information about the expertise and training provided to the outcomes assessors. 2. Harms: Please clarify how you will monitor for harms, and the types of adverse events collected. 3. Limitations: Please add further limitations to the discussion. Given you are conducting functional measures at ICU discharge and at 1 week post-ICU discharge, can you comment on the expected feasibility of conducting these outcomes. Margaret Herridge's recent Towards RECOVER study highlighted challenges conducting the 6MWT at 7 days post-ICU (Am J Respir Crit Care Med. 2016 Oct 1;194(7):831-844). 4. Pilot study? The additional file identifies this project as a "Preliminarily" randomized controlled trial – can the authors please clarify whether this is a pilot RCT or not? If this is a pilot RCT, please describe the criteria for progressing to a full RCT (e.g., BMJ 2016; 355 doi: https://doi.org/10.1136/bmj.i5239).
--	---

VERSION 1 – AUTHOR RESPONSE

Reviewer #1

General comment 1:

While much evidence has led to recommendations for early mobilization / early rehabilitation interventions for patients in the intensive care unit (ICU) there continues to be a need for evidence on interventions, and intervention dosing, for those patients. This study presents an effective proposal for evaluating one strategy for the management of exercise for these patients. The design of this study should be appropriate for meeting the objectives described on p. 5, lines 4 - 14, and that will expand the knowledge about managing this fragile patient population.

I found that this protocol article read well and made sense. This study should extend on what is already in the published literature, as was summarized on p. 4, lines 45 - 53. That information should prove important for critical care practitioners, especially those who provide rehabilitation services.

Response to General comment 1:

We appreciate these positive sentiments.

Comment 2: Note that the claim on p. 4, lines 45 - 48 will benefit from a citation.

Response 2: Two references have been added to the manuscript regarding the statement on p.4, lines 45-48. (page 4, Background and Rationale, 1st sentence of last paragraph)

Comment 3: My concern, as identified in the review checklist, is "4. Are the methods described sufficiently to allow the study to be repeated?" Rehabilitation interventions are notorious for being insufficiently described in the literature. The interventions in this protocol are described sufficiently for the study to be repeated except for the intensity of the exercise intervention. That is, my primary concern here is the planned "adjustment" of the ergometer "to facilitate optimal patient intensity." As described in the TIDieR checklist- "Tailoring: If the intervention was planned to be personalised, titrated or adapted, then describe what, why, when, and how" (from Hoffmann TC, et al. Better reporting of interventions: template for intervention description and replication [TIDieR] checklist and guide. *BMJ* 2014;348:g1687 doi: 10.1136/bmj.g1687, p. 6). It appears that this intervention will be titrated to the subjects' response. I recommend that elaboration on how intensity will be determined, and adjusted, will be informative for the provision of the intervention and for the replication of this study.

Response 3: Study Intervention section has been adjusted to improve clarity of the intervention to be delivered to enable replication of the study / intervention into clinical practice. Specific information regarding the utilisation of the BORG rate of perceived exertion and target value have been added to the manuscript. (page 7, Study Intervention, 11th and 12th sentences)

Comment 4: I also have concern about the exclusion criteria, as described in Table 1 (p. 6). I agree with the need for exclusion of subjects with a pre-existing condition that is likely to impair mobility / mobility assessment. It may improve clarity to change from examples for which subjects will be excluded (e.g., line 33) to specifics for which subjects will be excluded (that is, consider changing "e.g." to "i.e."). A similar concern applies to the descriptor for dire prognosis that is identified on line 43.

Response 4: The Exclusion criteria list in Table 1 has been updated. Each of the 'e.g.' written in have been edited to now be 'i.e.'

Comment 5: Beyond that my observations are rather petty, such as spelling errors (e.g., morphology, p. 8, line 39; and, p. 14, line 28) that will be resolved in the editing process.

Response 5: We have corrected the typographical errors on p.8. line 39 and p. 14, line 28.

General Comment 6: I found the overall design to be appropriate for a complex environment with at-risk and fragile subjects. The authors have described a reasonable strategy to enrol and to protect the subjects while engaging them in this research. The safety guidelines (table 2, p. 7 - 8) are consistent with standards for protecting patients participating in early rehabilitation in the ICU.

Thank you for this opportunity to review this proposal and I hope my review is informative.

General Response 6: The authors appreciate these comments which have enabled us to further improve the clarity and readability of this manuscript.

Reviewer #2

General comment 7: Nickels et al propose a phase IIb randomised controlled trial of in-bed cycling for the primary prevention of muscle wasting in critically ill patients. This addresses a highly relevant gap

in the literature. Whilst the study methodology is sound, I have a few comments that I hope will be taken in the spirit they are offered- to improve study design.

Response 7: We appreciate this constructive review.

Comment 8: The time has passed for unselected patients entering rehabilitation or prevention trials in critical care. Instead we are searching for more nuanced work. Publications from Herridge et al, Puthuchery et al and upcoming publications from the RECOVER trial demonstrate the variable responses of our patients to interventions, and I would suggest you either stratify patients for potential response, or adjust for baseline. This may involve retrospective HRQOL via Families, CPAT, functional histories (e.g. Ferrante JAMA int med 2015) or clinical frailty scoring. At least 2-3 trials have been skewed as result of not adjusting for baseline.

Response 8: The authors agree that the primary outcome should be adjusted for baseline and recognise the ability of adjusting for baseline to remove noise and give a clearer picture of a treatment's benefit. We can confirm that there is already scope within our analysis plan (using generalised linear mixed models) to adjust for baseline (if / when indicated). (page 12, Analysis, 2nd paragraph).

Comment 9: You have not stated that you will collect nutritional data- exercise and nutrition being intertwined in muscle physiology (see Bear Lancet Resp 2014). Given point 1 above (comment 6) I would collect data for NUTRIC scoring as this is likely to predict response to some extent.

Response 9: The authors agree that nutritional data and exercise are intertwined. We are collecting nutritional information (Pg. 9 line 16). The nutritional information collected will be calories and protein delivered. We are collecting sufficient data to enable a NUTRIC score to be calculated (without IL6) to enable further analysis. This information will be analysed in a Planned Secondary Analysis (Table 5, Pg. 12, Line 3) of the relationship between muscle wasting and participants nutritional intake whilst in ICU (page 13, Table 5, 6th planned secondary analyses).

Comment 10: Are femoral lines contraindicated? Hodgson has published consensus on mobilisation, and Parry's FES pilot study did not use this as an exclusion criteria.

Response 10: Femoral central lines are not contraindicated (footnote to Table 2 – Safety Guidelines). However, on advice of the treating intensivists at the participating hospital, patients with femoral lines greater in size than a central line (i.e. dialysis catheter, IABP, ECMO, or other lower limb arterial line) will not be cycled.

The rationale for this recommendation was the potential risk of a vascular injury with repetitive cycling motion in the limb with a large gauge vascular access was perceived by the clinical teams working in this ICU to be too high for patients with those types of femoral access to justify the inclusion of patients at this stage. However, the contralateral limb can be cycled (see footnote to Table 2).

Comment 11: How will the change in muscle mass be assessed statistically? This may work better with a RM-ANOVA as opposed to t-test.

Response 11: As outlined in the analysis plan, generalized linear mixed models (GLMM) will be used to analyse these data whilst accommodating repeated measures. A GLMM has similarities to a repeated measures ANOVA, but avoids the documented problems that RM ANOVA suffers when the data are not perfectly balanced – which is rare in human trials (see Diggle et al, Analysis of Longitudinal Data, 3rd Edition). The model is a regression model that will examine change from baseline with key predictors of treatment group to assess the effect of interest and baseline to adjust for differences between participants. A random intercept for each participant will adjust for within-

participant correlation. (page 12, Analysis, 2nd paragraph)

Comment 12: 20% missing data would be a worrying level- Imputation would not be appropriate at >10%

Response 12: Every effort will be made to minimise the amount of missing data and we will transparently report on drop-outs etc. to inform the reader about the nature of missing data. There will inevitably be some loss of data in this sample of acutely unwell patients. Multiple imputation can help reveal potential biases when 20% or more of the data are missing. For example, if more data were missing in the sickest patients with the poorest trajectories then a complete case analysis may over-estimate the treatment effect. An analysis that imputes missing data is therefore an important sensitivity analysis to confirm the results using the complete data.

Reviewer #3

General comment 13: Nickels et al. describe a research protocol for in-bed cycling with critically ill patients. It is a single-center, parallel group randomized trial with blinded outcome measures. Patients will be randomized to 30 minutes of cycling and routine physiotherapy interventions or routine physiotherapy interventions alone for up to 28 days, with a minimum of 5 cycling sessions whilst in hospital (ICU and acute ward combined). The primary outcome is change in rectus femoris cross-sectional area compared to baseline at 10 days post-study enrolment. Secondary outcomes include strength (MRC sum score, hand grip), function (ICU mobility scale, functional status score for ICU, 6-minute walk test), cognition (CAM-ICU up to 10 days post-enrollment), quality of life (EQ5DL 10 days post-enrolment, 3 and 6 months), ICU and hospital LOS, acute discharge destination, and mortality. Congratulations to the investigators on initiating an early randomized study of in-bed cycling with critically ill patients. We need more rigorous research in this field, and it is great to have others studying interventions to help our ICU survivors. The authors include some strong design features to ensure intervention exposure (e.g., minimum 5 cycling sessions), patient acceptability, and use common ICU outcomes. Given this study is already enrolling, I will focus on areas for clarification and better research reporting. Thanks for including the SPIRIT checklist.

Response 13: We appreciate these kind words, and the constructive review.

Comment 14:

Main issues 1. Primary outcome – The primary outcome is change in rectus femoris quadriceps cross sectional area measured at 10 days post-enrollment compared to baseline. The authors will enroll 68 patients to demonstrate a 2.9% difference in change between groups.

a. Rationale for primary outcome:

i. More information to justify the choice of this outcome, rather than a functional or patient-centered outcome is needed, since completed or ongoing trials reported outcomes of the 6MWT or feasibility. What is the clinical relevance of muscle cross sectional area, rather than function?

Response 14: Further justification has now been added to the Introduction to provide context for the selection of ultrasound as the primary outcome measure.

This study builds on the RCT by Burtin et al 2009 that found improved 6MWT distance at hospital discharge for those patients who participated in in-bed cycling, and the Acute Skeletal Muscle Wasting in Critical Illness study by Puthuchery et al. 2013. A limitation reported in the Burtin et al 2009 study was the absence of sonography or muscle biopsy measures that may assist to explain the improved physical outcomes of those that completed in-bed cycling. The study by Puthuchery et al. 2013 provided important objective data regarding the rate of skeletal muscle atrophy measured by ultrasound. Hence, this study will provide important new information building on this prior work, while also reporting on functional outcomes.

This trial will be an important step forward, and we have now added the additional justification for use

of muscle cross-sectional area as the primary outcome in this trial. (page 4, Background and Rationale, 9th sentence)

Comment 15: a. Rationale for primary outcome:

ii. Information about the rationale for the timing of this outcome (i.e., 10 days) is needed.

Response 15: Further information within the Outcomes section manuscript has been provided. We have added, 'Day 10 post study enrolment has been chosen to enable comparison with previously published data on acute skeletal muscle wasting in patients with critical illness and to reflect the timeframe of observed muscle wasting 1'. (page 9, Outcomes, 5th sentence)

Comment 16: a. Rationale for primary outcome:

iii. Please clarify the direction of change (e.g., the cycling group will have less change in CSA than usual care PT).

Response 16: Clarification within the Outcomes section manuscript has been made. We have added a sentence, 'The authors anticipate that less rectus femoris CSA atrophy will be observed among the in-bed cycling group.' (page 9, Outcomes, 4th sentence)

Comment 17: a. Rationale for primary outcome:

iv. A 2.9% difference in change score between groups is small. What is the expected precision of the measure between groups (i.e., 95% confidence interval)? The study cited to support the sample size calculation reported a mean change of -17.7% (-20.9% to -4.8%), representing a wide confidence interval for the control group.

Response 17: As an indicator of the assumed variation of difference in CSA% our calculation assumed a standard deviation of 6% (and a correlation between repeated measures of 0.5). We can see how this was confusing having listed the estimate of data variation for the calculation as a standard deviation, then listing a confidence interval range from a prior study (i.e., normal 95% CI range would \sim mean + and - 1.96 x SD). Given this reference to the prior study is somewhat confusing we have now removed it. We have also acknowledged some limitations in the data available to inform the sample size estimate within the sample size section of the manuscript. (page 10, Sample Size, last sentence).

Comment 18: a. Rationale for primary outcome:

v. Please clarify how many assessments are required per group (you will recruit 68 to account for 20% drop out rate). How does this sample size calculation account for ICU mortality?:

Response 18: Clarification within the Sample Size section of the manuscript has been made to specifically make reference to 68 participants being required to account for up to a maximum of 20% drop out rate including mortality (i.e., the calculation assumes that the loss of data due to mortality is included in that 20%). We have also clarified what this represented in terms of participants in each group (approximately n=27) required to complete the trial. (page 10, Sample Size section)

Comment 19:

b. Assessment of primary outcome: Please provide more information about the expertise and training provided to the blinded outcome assessor

Response 19: Clarification within the Data Collection section manuscript has been made. The description 'Registered post-graduate trained sonographers with experience in musculoskeletal sonography' has been added to the data collection section of the manuscript. Furthermore, we have also clarified that the sonographer assessors met with each other and the study coordinator prior to

the commencement of the trial to ensure assessment procedures were consistent between the outcome assessors. (page 11, Data Collection, 3rd paragraph)

Comment 20: Missed outcomes: On page 11, line 36, the authors report that if sonographers do not record the primary outcome, then a physiotherapist will use thigh circumference measures to impute missing outcome data. Please provide rationale for this statement, given poor psychometrics of thigh circumference measures (Parry et al., *Intensive Care Med* (2015) 41:744–762). Alternatively, would the authors consider adding a time window to conduct the 10-day ultrasound measure (e.g., +/- 1 day) and conduct a sensitivity analysis to see if there is a difference by measurement timing?

Response 20: We believe this line of text has inadvertently been misread and incorrectly transposed by the reviewer. There was no statement indicating that a physiotherapist will use thigh circumference as a measure. We have now clarified in the text that the physiotherapist experienced in musculoskeletal ultrasound will take a measurement using ultrasound as if a sonographer cannot complete their assessment within the ± 1 day window (e.g., due to unplanned staff absence). We have also added that 'the scan will be performed within a day (± 1 day) of the designated time-point if it is unable to be completed on the scheduled day (and this will be recorded for inclusion in analyses if necessary). Whilst the information from the physiotherapist will not be an exact substitute, it will contain some information that will be useful for multiple imputation if necessary. (revisions have been made on page 12, Analysis, 3rd paragraph)

Comment 21: 2. Study intervention: Some elements of the cycling intervention are not clear. Can the authors please clarify the following:

a. At the time of enrolment, do patients need to meet the safety criteria on page 7, table 2? How will you manage patients enrolled in the study, randomized to the cycling arm, and don't receive the treatment?

Response 21: Clarification provided within the Participants section of the manuscript, 'Participants can be recruited into the study and baseline sonography measures performed if a patient does not currently meet the in-bed cycling safety criteria in Table 2.' (page 6, Participants, 3rd sentence of 1st paragraph)

Comment 22: 2. Study intervention:

b. Will this study prioritize routine physiotherapy interventions over in-bed cycling (see page 6, line 56)? The authors report "safety guidelines will be used to determine if the intervention group patients are able to complete an additional daily 30 minute...."

Response 22: Usual care physiotherapy will be provided by the usual clinical team and prioritised over in-bed cycling. In bed cycling will be provided in addition to usual care (within the safety guidelines). Clarification has been made to the study intervention section of the manuscript, 'Usual care physiotherapy interventions will be provided to all patients (prioritised ahead the in-bed cycling intervention if necessary).' (page 7, Study Intervention, 3rd sentence)

Comment 23: 2. Study intervention:

c. Please justify why the initial cycling rate is 20 RPM, and the rationale for adding resistance.

Response 23: Clarification has been made to the Study Intervention section of the manuscript, 'this is the default passive speed of the bedside cycle ergometer'. A cycling rate of 20rpm was the speed used in the cycle ergometry study by Burtin et. al. 2009, that reported improved physical function measured by 6MWT distance at hospital discharge. To the authors knowledge there are no other published studies that have reported functional outcomes using different cycling rates (cadence). The Study Intervention section of the manuscript that discusses the rationale behind increasing

resistance has been re-written to improve clarity and additional references have also been incorporated (Borg 1990 and Berney et al. 2012). (page 7, Study Intervention, 9th, 11th and 12th sentences)

Comment 24: 2. Study intervention:

d. Please clarify the number of days per week cycling is offered as part of the study (e.g., 5 days per week, 7 days per week, other). Is this frequency of intervention similar to current ICU PT scheduling?

Response 24: The Study Intervention of the manuscript has been modified and edited to improve clarity about sessions per week and to confirm that the frequency of intervention is up to 7 days per week, which is similar to ICU physiotherapy usual care. (page 7-8, Study Intervention, bottom of page 7 to top of page 8)

Comment 25: 2. Study intervention:

e. Given additional safety criteria for active cycling, how will the authors manage patients who transition from passive to active cycling in the same session and have an active exercise exclusion?

Response 25: The Study Intervention section of the manuscript has the following information added to improve clarity, 'If a patient unexpectedly commences active cycling the additional active in-bed cycling safety criteria will be then be applied. If the patient is deemed unsuitable to continue active in-bed cycling they will be asked to resume passive cycling. If the patient continues to actively cycle the session will be ceased by the treating clinician.' (page 7, Study Intervention, 13th, 14th and 15th sentences)

Comment 26: 2. Study intervention:

f. Please provide more information about the expertise and training provided to the interventional physiotherapist for cycling.

Response 26: The Study Intervention section of the manuscript has had the following information added to improve clarity, 'A lead physiotherapist with over 10 years of experience in rehabilitative exercise with critically ill patients will be responsible for conducting the in-bed cycling sessions. He has been trained by industry cycle ergometry representatives and has over 5 years of experience conducting in-bed cycling sessions with critically ill patients. Experienced ICU physiotherapists trained by the lead physiotherapist in conducting in-bed cycling sessions may conduct the in-bed cycling sessions if the lead physiotherapist is unavailable.' (page 7, Study Intervention, 5th, 6th and 7th sentences)

Comment 27: 3. Control group: How will authors document the frequency, intensity, type, and duration of physiotherapy interventions in the control group

Response 27: Prior to the present trial, we conducted a large audit of usual care at this facility (>3000 cases over a 5 year period) which we anticipate will be published prior to the reporting of this trial that we will be able to refer to when describing usual care at the participating facility. We do not anticipate that the number of routine physiotherapy interventions will be affected (as they are delivered by the usual treating clinical teams independent of the study and prioritised over in-bed cycling) and the study design is a randomised controlled trial, so it is anticipated that usual care physiotherapy interventions will be similar across groups. Within the finite resources we have to complete this study, we are not planning to prospectively document the frequency, intensity, type and duration of usual care interventions for each patient as part of the trial measurements. We have ensured this has now been acknowledged in the discussion. (page 15, Discussion, last paragraph)

Comment 28: Other issues:

1. Secondary outcomes: Please provide more information about the expertise and training provided to the outcomes assessors.

Response 28: The Data Collection section of the manuscript has the following information added to improve clarity, 'Assessment of muscle strength and function will be completed by cardiorespiratory physiotherapy assessors with a minimum of 3 years' experience at ICU discharge and 7-days post ICU discharge. These physiotherapists will be blinded to group allocation. The same physiotherapists will also assess 6-minute walk test distance 7-days following ICU discharge. The study physiotherapists were trained by the lead investigator in the standardised assessment of the study outcome measures prior to the commencement of the study.' (page 11, Data Collection, last paragraph)

Comment 29: Other issues:

2. Harms: Please clarify how you will monitor for harms, and the types of adverse events collected.

Response 29: The Trial Management section of the manuscript has the following information added to improve clarity, 'During in-bed cycling sessions participants will be monitored for adverse events that will be recorded on the session data collection form. Adverse events that are being monitored for are; line or airway dislodgement, increase in ventilatory support that persist greater than five minutes post exercise (e.g. increase PEEP or FiO₂), blood oxygen desaturation less than 88% for more than one minute, increase in vasoactive or pain relief medication greater than 5mcg/min, increase in systolic blood pressure greater than 180mmHg for more than two minutes, increase in heart rate greater than 140 for more than two minutes, decrease in mean arterial blood pressure less than 60mmHg for greater than two minutes and decrease in heart rate less than 50 beats per minutes for more than two minutes.' (page 13-14, Trial Management, last paragraph).

Comment 30: Other issues:

3. Limitations: Please add further limitations to the discussion. Given you are conducting functional measures at ICU discharge and at 1 week post-ICU discharge, can you comment on the expected feasibility of conducting these outcomes. Margaret Herridge's recent towards RECOVER study highlighted challenges conducting the 6MWT at 7 days post-ICU (Am J Respir Crit Care Med. 2016 Oct 1;194(7):831-844).

Response 30: The Discussion section of the manuscript has the following limitation added, 'A potential limitation of the study may be difficulty for a small proportion of patients to complete functional assessment measures at ICU discharge and at one week post-ICU discharge; for example, patients who are profoundly weak or present with acute cognitive dysfunction may not be able to complete the functional assessments at the scheduled times.' (page 15, Discussion, last paragraph)

Comment 31: Other issues:

4. Pilot study? The additional file identifies this project as a "Preliminarily" randomized controlled trial – can the authors please clarify whether this is a pilot RCT or not? If this is a pilot RCT, please describe the criteria for progressing to a full RCT (e.g., BMJ 2016; 355 doi: <https://doi.org/10.1136/bmj.i5239>).

Response 31: Clarification has been added to the Discussion section that this study is a 'Phase IIb' RCT. The following information regarding progressing to a Phase III RCT has also been added, 'At

the completion of the study further consideration will be given as to whether a definitive larger-scale Phase III trial is warranted. This will include consideration of the evidence of effect on primary and secondary outcomes, participant recruitment and adherence to the in-bed cycling protocol and rate, as well as acceptability of the intervention.' (page 15, Discussion, 2nd paragraph)

Overall Summary:

The authors appreciate the suggestions and comments that have been addressed. The manuscript has been edited to improve its' readability. Clarifications have also been made that enable others to replicate the study and in-bed cycling intervention.

VERSION 2 – REVIEW

REVIEWER	James M Smith Utica College, USA
REVIEW RETURNED	13-Jul-2017

GENERAL COMMENTS	The authors have addressed the concerns raised by the reviewers. I have no additional recommendations or concerns.
--

REVIEWER	Zudin Puthuchery Royal Brompton Hospital, United Kingdom
REVIEW RETURNED	09-Jul-2017

GENERAL COMMENTS	The authors have addressed the reviewers comments in detail and adequately, though the manuscript does not explicitly state that they will adjust for baseline status...
--

VERSION 2 – AUTHOR RESPONSE

The authors appreciate the reviewers' suggestion regarding explicitly stating whether we will adjust for baseline status.

We have modified the manuscript to be explicit that we are utilising a statistical model that enables adjustment for baseline status. We have also added a reference that supports adjusting for baseline status when observed baseline value influences the exposure. 'Linear regressions adjusted for baseline level should be preferred to unadjusted linear regression analyses'. As participants will present with different baseline values of muscle cross-sectional area and potentially respond differently to exposure to the intervention the ability to adjust for baseline status is important. The reference cited is:

Lepage B, Lamy S, Dedieu D, et al. Estimating the Causal Effect of an Exposure on Change from Baseline Using Directed Acyclic Graphs and Path Analysis. *Epidemiology* 2015;26(1):122-29.